# Limitations in the Grain Boundary Processing of the Recycled HDDR Nd-Fe-B System

**DOI:** 10.3390/ma13163528

**Published:** 2020-08-10

**Authors:** Awais Ikram, Muhammad Awais, Richard Sheridan, Allan Walton, Spomenka Kobe, Franci Pušavec, Kristina Žužek Rožman

**Affiliations:** 1Faculty of Mechanical Engineering, University of Ljubljana, Aškerčeva cesta 6, SI-1000 Ljubljana, Slovenia; franci.pusavec@fs.uni-lj.si; 2Department for Nanostructured Materials, Jožef Stefan Institute, Jamova 39, SI-1000 Ljubljana, Slovenia; spomenka.kobe@ijs.si (S.K.); tina.zuzek@ijs.si (K.Ž.R.); 3Jožef Stefan International Postgraduate School, Jamova 39, SI-1000 Ljubljana, Slovenia; 4School of Metallurgy and Materials, University of Birmingham, Edgbaston, Birmingham B15 2TT, UK; m.awais@bham.ac.uk (M.A.); r.s.sheridan.1@bham.ac.uk (R.S.); a.walton@bham.ac.uk (A.W.)

**Keywords:** rare earth permanent magnets, HDDR Nd_2_Fe_14_B, recycling, spark plasma sintering, grain boundary diffusion processing (GBDP), high coercivity

## Abstract

Fully dense spark plasma sintered recycled and fresh HDDR Nd-Fe-B nanocrystalline bulk magnets were processed by surface grain boundary diffusion (GBD) treatment to further augment the coercivity and investigate the underlying diffusion mechanism. The fully dense SPS processed HDDR based magnets were placed in a crucible with varying the eutectic alloys Pr_68_Cu_32_ and Dy_70_Cu_30_ at 2–20 wt. % as direct diffusion source above the ternary transition temperature for GBD processing followed by secondary annealing. The changes in mass gain was analyzed and weighted against the magnetic properties. For the recycled magnet, the coercivity (*H_Ci_*) values obtained after optimal GBDP yielded ~60% higher than the starting recycled HDDR powder and 17.5% higher than the SPS-ed processed magnets. The fresh MF-15P HDDR Nd-Fe-B based magnets gained 25–36% higher coercivities with Pr-Cu GBDP. The FEG-SEM investigation provided insight on the diffusion depth and EDXS analysis indicated the changes in matrix and intergranular phase composition within the diffusion zone. The mechanism of surface to grain boundary diffusion and the limitations to thorough grain boundary diffusion in the HDDR Nd-Fe-B based bulk magnets were detailed in this study.

## 1. Introduction

The Nd-Fe-B based rare-earth permanent magnets (REPMs) possess great significance for microelectronics, data storage, electric motors and medical devices [1]. The grain size refinement has been theorized to improve the coercivity (*H_C_*), i.e. resistance to demagnetization in REPMs [2]. The hydrogenation-disproportionation-desorption-recombination (HDDR) is a well-established and greener route for developing anisotropic ultrafine grains (~400 nm) with preferential easy-axis orientation [3]. The overall surge in production volume and application demand has necessitated the utilization of recycled REPMs into the industrial feedstocks [4,5]. The effectiveness of hydrogen gas towards decrepitating (HD) and disproportionation (HDDR) of the end-of-life rare-earth (RE) scrap has convincing been proven in previous studies [6,7,8,9,10,11,12]. Contrary to the theoretical models, the HDDR Nd-Fe-B system lacks high coercivity as a translation of high magnetocrystalline anisotropy (*µ_0_H_a_*) ~ 7.2 T because of crystal structure inhomogeneities, grain morphology, surface defects, oxidation, nonferromagnetic grain boundaries and localized exchange interactions at the grain interfaces, which restrict their usability [13,14,15,16,17]. Therefore, rather confining the potential application of the recycled HDDR Nd-Fe-B powders to polymer bonded magnets only [15,18,19,20], we have recently demonstrated Spark Plasma Sintering (SPS) as a convenient method to practically fabricate bulk magnets with magnetic properties at par with the end-of-life sintered magnets [21,22,23] or commercial grade HDDR Nd-Fe-B powder [24]. Several researchers have further explored the possibilities of improving the coercivity in the HDDR Nd-Fe-B system, with alloying additions [25,26,27,28,29], by mechanical milling [30,31,32,33,34] and via tailoring the disproportionation-desorption-recombination parameters [10,11,35,36,37,38,39,40,41]. 

More pragmatic approach has been the application of grain boundary diffusion (GBD) treatment on the HDDR Nd-Fe-B powders [36,42,43]. In case of the bulk magnets obtained via hot-pressing the HDDR Nd-Fe-B powder Song et al. [43] achieved *µ*_0_*H_C_* = 1.55 T (*H_Ci_* = 1230 kA/m) by eutectic Pr-Cu GBD treatment. Another versatile method applied to the bulk sintered magnets from the HDDR Nd-Fe-B powder has demonstrated that by doping with the rare-earth (RE) fluoride (DyF_3_) and controlled heat treatment, an improvement of up to 70% in coercivity (*H_Ci_*) can be accomplished [44]. As compared to the sintered magnets from microcrystalline jet-milled powders and the nanocrystalline hot-deformed melt-spun ribbons, the GBD process has not been extensively researched on the HDDR Nd-Fe-B system. Moreover, a step ahead, very limited published data is available on the bulk HDDR Nd-Fe-B magnets, let aside GBD processing. When comparing the existing scientific reports, the underlying mechanism of diffusion has been not interpreted either for the HDDR Nd-Fe-B system [36,42], and sparsely interlinked with hot deformed nanocrystalline and sintered Nd-Fe-B magnets [45,46,47,48]. Citing the GBDP of the HDDR Nd-Fe-B powder, the loose powder particles have a remarkably higher surface area as compared to bulk sintered magnets and therefore, the suggested improvement in coercivity is only due to the widening of intergranular boundaries along the Nd_2_Fe_14_B grains which would effectively decouple the grains with a higher concentration of non-ferromagnetic elements in this spacer phase [29,49]. Accordingly, the diffusion mechanism is relatively straightforward for the loose HDDR powder particles undergoing GBD treatment, such that the liquid phase engulfs all the particles when 5–30 wt. % RE-rich alloys are added and the short-range diffusion causes widening of grain boundary regions within individual particles [36,42]. On the contrary, when compacted to full density, the diffusion mechanism in the HDDR Nd-Fe-B system is not the same as the loose powder particles and must vary with the GBD processing parameters. The previous report on the hot deformed HDDR Nd-Fe-B bulk magnets does not compliment on the diffusion mechanism either and relates more to the improvement in magnetic properties after GBDP with hyper-eutectic Pr_82_Cu_18_ alloy [43]. The recycled HDDR powder when compacted with SPS contains particle boundaries within which the RE-rich phase is non-uniformly distributed before annealing. The RE-rich phase gets transported from the interparticle region to the grain boundaries during the annealing and as a result, the coercivity reportedly increased [22]. This non-uniform distribution of Nd-rich phase in the particle boundaries happened under intense uniaxial pressure in the SPS such that liquid phase was squeezed out of grain boundaries to the interparticle junctions, and as a result, the coercivity dropped because of localized exchange interactions within the adjoining Nd_2_Fe_14_B grains. During annealing, this liquid phase is partially transported back towards the grain boundaries surrounding ~400 nm sized Nd_2_Fe_14_B matrix grains from these interparticle boundaries, as the system is equilibrated. When evaluated based on particle size, we were able to distinguish that for the smaller particle (<100 µm) there is a loss in total grain boundary area which also consequently resulted in a higher degree of oxidation as the liquid phase was squeezed out to the particle boundaries during SPS, where the bulk of hcp-Nd_2_O_3_ phase transformation took place [21]. The capillary transport was theorized and confirmed during the local doping and grain boundary engineering of bulk HDDR Nd-Fe-B magnets with DyF_3_ nanoparticles [44].

Evidently the insufficient understanding of the demagnetization (or coercivity) mechanism, the obvious effect of RE-rich interfaces—the grain boundaries as well as the surface diffusion kinetics in the HDDR Nd-Fe-B system potentially constrain their commercial applicability [2,14,42,49,50,51,52,53].

The rudimentary aim of this scientific briefing is to highlight the potential challenges in grain boundary diffusion processing (GBDP) of the recycled HDDR Nd-Fe-B based fully dense bulk magnets in comparison to the commercial (Aichi MF-15P) HDDR powder with respect to the magnetic properties evolution after the GBD treatment with Pr-Cu and Dy-Cu eutectic alloys. In case of direct diffusion source and surface GBDP, the uniform and thorough dispersal of the RE-rich liquid phase to the grain boundaries under enhanced capillary transport was observed to be constricted in the bulk HDDR Nd-Fe-B magnets and this phenomenon has not been previously identified or reported elsewhere. Moreover, the diffusion depth after the thermal treatments and the effect of processing temperatures have been investigated in this report. This research also highlights the surface diffusion mechanism in the recycled HDDR Nd-Fe-B system from direct diffusion source, which has not been previously reported and interlink has been devised from the context of mass gain with the different GBDP parameters. The fundamental challenge regarding the diffusion depth limitations in the HDDR Nd-Fe-B bulk magnets has been investigated and proposed due to partial capillary diffusion confined due to the presence of complex intergranular oxides with the aid of SEM and EDXS analysis. Apart from the complicated diffusion mechanism in the HDDR Nd-Fe-B bulk magnets, the asymmetrical transformation of (Pr,Nd)_2_Fe_14_B phase facets along with the interconnected Pr-Cu rich liquid phase and the (Pr,Nd)_X_CuO_Y_ complexes at the intergranular junction have been suggested in correlation with the different GBDP parameters.

## 2. Contribution to the Field Statement

The majority of work on GBDP of Nd-Fe-B system accounts to sintered magnets from the microcrystalline powders and the nanocrystalline melt-spun ribbons, either in form of milled powder or hot-deformed magnets. On the contrary, citable literature on the Hydrogenation-Disproportionation-Desorption-Recombination (HDDR) Nd-Fe-B system is scarce such that these two nanocrystalline Nd-Fe-B systems are radically different and the diffusion treatment mechanism is relatively sparsely understood for the HDDR system. Importantly, the direct magnet recycling philosophy utilizes processing magnetic scrap with hydrogen, like in hydrogen decrepitation (HD) or HDDR, with latter technique extensively applicable for producing anisotropic ≤400 nm nanostructured Nd_2_Fe_14_B grains and high coercivity bonded magnets. The GBDP on the HDDR powders have been reported [36,42] citing coercivity improvement but without the explanation of the diffusion mechanism in comparison to the sintered magnets or the nanocrystalline melt-spun ribbons. A recent study suggested application of Pr-Cu low melting alloy to HDDR powder treated with hot deformation, improving the coercivity from 1065 kA/m to 1232 kA/m [43]. However, the bulk diffusion depths and the mechanism of the particle to grain boundary diffusion were still lacking.

The concurrent research work addresses the diffusion mechanism in the recycled HDDR Nd-Fe-B system, which has never been reported before. The details are presented in the context of mass gain with diffusion processing parameters and the diffusion depth limitations in the system due to limited capillary channelling and presence of complex intergranular oxides with the aid of SEM and EDXS analysis. Beyond the complications of constricted diffusion depth in the dense HDDR Nd-Fe-B bulk magnets, the asymmetrical formation of (Pr,Nd)_2_Fe_14_B phase along the facets rich with Pr-Cu containing liquid phase and (Pr,Nd)_X_CuO_Y_ phase transformation at the intergranular junction under different processing conditions have been identified. The magnetic properties as a comparison were also analysed after the diffusion treatment on the commercial HDDR Nd-Fe-B based bulk magnets, which resulted in ~25% coercivity grain whereas the reprocessed magnets yielded ~60% higher coercivity over the starting recycled HDDR powder. This brief report also indicates further action plans for the future on GBDP of the HDDR Nd-Fe-B system to tackle the limitations in diffusion depth and augment the coercivity beyond the current state of the art values.

## 3. Materials and Methods

The cylindrical/disk-shaped bulk SPS reprocessed magnets of diameter 9.5 mm and height 3 mm were prepared from the recycled HDDR Nd-Fe-B powder of nominal atomic composition: Nd_13.4_Dy_0.67_Fe_78.6_B_6.19_Nb_0.43_Al_0.72_ (and in the mass ratio: Nd_29.46_Dy_1.66_Fe_66.94_B_1.02_Nb_0.65_Al_0.30_). The HDDR processing on the end-of-life magnets with this composition has been well documented by Sheridan et al. [10,54]. Sequentially, the fabrication of bulk sintered magnets from the recycled HDDR Nd-Fe-B powder has also been thoroughly elaborated previously by Ikram et al. [21,22], including the physical properties like density (7.57 g/cm^3^), oxygen content (4800 ppm) and anisotropic powder particles in a wide distribution from 30–700 µm (average size ~220 µm). As a comparison, the commercial grade Aichi’s Magfine MF-15P HDDR Nd-Fe-B anisotropic powder (Aichi-ken, Japan), with average particle size 120 µm in a narrow size distribution was also consolidated with the similar SPS processing conditions [23] to replicate the GBD effect on fresh material. The binary eutectic Pr_68_Cu_32_ and Dy_70_Cu_30_ alloy ribbons were prepared through arc-melting 10 g of elements in stoichiometric compositions (MAM-1 Arc Melter, Edmund Bühler, Bodelshausen, Germany). The Pr-Cu and Dy-Cu alloys were homogenized by 5 arc-melting passes and cooled to room temperature before grindings off the surfaces with 500 grit SiC papers. This was followed by vacuum induction melting (>10^−4^ mbar) and subsequent melt spinning on a 200 mm copper wheel with rotational velocity of 30 m/s in argon atmosphere (MSP-10 SC Melt Spinner, Edmund Bühler, Bodelshausen, Germany). The alloy ribbons (several mm thickness, without further comminution) were placed within a ceramic crucible as direct diffusion source on the top and bottom of surface cleaned bulk magnets (thermally annealed at 750 °C for 1 h) [21]. The loaded crucible was placed in a horizontal tube furnace (Carbolite Gero Limited, Hope Valley, UK) for the GBD processing in high vacuum (>10^−5^ mbar) with a heating rate of 50 °C/min. The GBD treatment was performed at 900 °C for 3 h with secondary annealing at 500 °C for additional 3 h in the same tube furnace setup (vacuum and heating rate), which is above the ternary transition temperature [22] to accelerate the surface diffusivity. The magnetic measurements on bulk GBDP samples were performed at room temperature on a permeameter (Magnet-Physik Dr. Steingroever, Cologne, Germany) with demagnetizing fields up to 2 T. For further characterization, the samples were thermally demagnetized at 400 °C for 15 min in the tube furnace (vacuum > 10^−5^ mbar). Samples were sliced into half by low speed diamond saw at 300 rpm and fluxed with isopropanol (IsoMet™ Precision Cutter, Buehler, IL, USA). Later, these demagnetized GBDP samples were grinded by 500, 1000 and 2400 grit size SiC papers. Successively, the polishing was done with 0.25 µm diamond paste slurry on a velvet cloth at 200 rpm. The microstructural investigation was accomplished with JEOL 7600F (Field Emission Scanning Electron Microscope–JEOL Ltd., Tokyo, Japan) with an electron energy dispersive X-ray spectroscopy (EDXS) analyzer and a 20 mm^2^ Oxford INCA 350 detector (Oxford Instruments, High Wycombe, UK) for compositional/elemental analysis, performed at 20 keV accelerating voltage.

## 4. Results and Discussion

The recycled and fresh Magfine MF-15 HDDR Nd-Fe-B magnets were developed by SPS sintering operation at 750 °C with holding time of 1 minute, and further complemented by thermal treatment at 750 °C for 1 h. The magnetic properties of HDDR powder and annealed bulk magnets were measured prior to the GBD treatment with eutectic Pr-Cu and Dy-Cu alloy ribbons and summarized in Table 1 [22].

### 4.1. Dy-Cu Grain Boundary Diffusion Treatment

The grain boundary diffusion (GBD) treatment parameters were contemplated from a similar study on Nd-Fe-B melt-spun ribbons by Bao et al. [46] in which Dy_70_Cu_30_ and Pr_68_Cu_32_ were utilized. The starting bulk magnets had a relative density > 99% after the thermal treatment, therefore they can be classified as fully dense magnets before GBDP. The eutectic melting point for Dy_70_Cu_30_ was reported as 790 °C. In order to promote eutectic alloy ribbons fluidity and uniform melting on the surfaces of bulk magnets, the primary GBDP temperature was retained at 900 °C for 3 h. Secondary annealing at 500 °C for 3 h was opted to relax the thermal strains formed in the bulk magnets during GBDP. The mass gain (up to 3 digits precision) was analyzed to overview how much the molten species have diffused into the magnets after GBDP. The starting magnets ranged from 2.600–2.800 g in nominal masses after the initial grinding was performed to remove carbon layer off the SPS-ed specimen. The actual mass gain in reprocessed magnets (RMs) was as follows: for 2 wt. % sample approx. 0.019 g (1.94%), 10 wt. % sample approx. 0.068 g (2.57%) and 20 wt. % sample approx. 0.0911 g (3.25%).

The augmentation of the magnetic properties is clearly illustrated in Figure 1. The coercivity (*H_Ci_*) of the starting bulk magnets (1148 kA/m) increased to a modest value of 1220 kA/m with 2.wt. % Dy-Cu eutectic alloy addition at the first stage of GBDP and slightly more to 1250 kA/m after secondary annealing at 500 °C. In case of 10 wt. % Dy-Cu alloy added as the diffusion source, *H_Ci_* increased from 1141 kA/m to 1216 kA/m at 900 °C and finally to 1257 kA/m with the secondary annealing. For 20 wt. % there was a subsequent increase in coercivity from 1149 kA/m to 1226 kA/m (at 900 °C) and 1287 kA/m (at 500 °C), as shown in Figure 1a. Whereas the remanence (*B_r_*) of the starting bulk magnets prior to GBDP was uniformly 0.81 T and it consistently dropped more for primary diffusion processing at 900 °C with a minor recovery after secondary annealing at 500 °C. Earlier investigation on hot-deformed HDDR Nd-Fe-B magnets suggested that the thermal treatment above the ternary transformation point introduces higher degree of alignment (texturing) and the subsequent relaxation in processing strains introduced by grain boundary phase [23]. This factor may be at play here such that the RE-enriched liquid phase at 900 °C realigns the HDDR particles slightly, since the melt-flows preferentially along the c-axis, resulting in a minor improvement in remanence after annealing [22]. The overall the drop in the *B_r_* due to a higher weight fraction of Dy-Cu alloy after the GBD treatment has been reported due to the antiferromagnetic coupling introduced by partial substitution of Nd with the Dy atoms [55] at the surface of the nanocrystalline Nd_2_Fe_14_B matrix grains during the primary diffusion treatment at elevated temperatures, above the binary eutectic 790 °C for Dy_70_Cu_30_ and ternary transition temperature of Nd-Fe-B alloys (665 °C) [44,46]. This 20 wt. % addition of Dy-Cu eutectic ribbons caused approximately 55% improvement in H_Ci_ over the starting recycled HDDR Nd-Fe-B powder and 12% better demagnetization resistance over the optimally SPS reprocessed bulk magnets. Comparing the gain in masses after GBDP, the surface of RMs still contained brazed and non-diffused species on the top and bottom, so the surface was grinded to the original height of the bulk magnet for magnetic measurements. This partial mass gain, besides very slight decline in the *B_r_* (<0.1 T) for the diffusing Dy indicates that the Dy-Cu alloy ribbons did not melt properly under the 900 °C GBDP conditions, owing to their high eutectic temperature which caused very limited H_Ci_ improvement in RMs.

### 4.2. Pr-Cu Grain Boundary Diffusion Treatment

The eutectic melting regime for Pr_68_Cu_32_ based melt-spun ribbons was identified at 472 °C, in which the authors also suggested Pr-Cu alloys were found more effective than Dy-Cu system to augment the coercivity, without conversely affecting the remanent magnetization [46]. The change in relative density was insignificant after the GBDP of reprocessed magnets (RMs) and values were consistently higher than 7.56 g/cm^3^ after primary diffusion treatment and secondary annealing. The actual mass gain in Pr-Cu GBDP reprocessed magnets (RMs) was higher as compared to the Dy-Cu based RMs, with measurements detailed as follows: for 2 wt. % sample approx. 0.055 g (2%), 10 wt. % sample approx. 0.105 g (3.87%) and 20 wt. % sample approx. 0.138 g (4.96%). Apparently the Pr-Cu alloy diffused from the surface in 2 wt. % condition, the brazed species were still present in 10 and 20 wt. % RM samples which were finely grinded to the preceding dimensions.

The magnetic properties prior to and after the GBDP are illustrated in Figure 2, indicating a more profound effect of Pr-Cu alloy in improving the coercivity without appreciable reduction in the remanent magnetization. For 2 wt. % Pr-Cu ribbons, a minor improvement in *H_Ci_* = 1221 kA/m was observed (RM *H_Ci_* = 1157 kA/m) during primary diffusion which further increased to 1283 kA/m after annealing, as shown in Figure 2a. This value corresponds to similar *H_Ci_* improvement as possible with 20 wt. % Dy-Cu alloy (actual 3.25% mass gain). The *B_r_* improved slightly to 0.82 T over the original starting RM (0.8 T) after 2 wt. % GBDP, which indicates Pr diffusion is overall more efficient in augmenting the *H_Ci_* without negatively impacting the *B_r_* as in case of Dy diffusion. The *B_r_* dropped slightly to 0.78 T (10 wt. %) and 0.75 T (20 wt. %) Pr-Cu GBDP, as shown in Figure 2b indicating the dilution of the hard-magnetic Nd_2_Fe_14_B phase with the non-ferromagnetic species diffusing inwards from the surface [45,46,47,56,57]. Consequently, with more mass gained by 10 and 20 wt. % Pr-Cu samples, it is plausible that the remanent magnetization is expected to get reduced since these eutectic alloys are compensating and becoming part of the intergranular phase, making it richer with the REEs content. With more intergranular phase and unchanged volume fraction of the ferromagnetic phases, we can expect the improvement in coercivity but only at the cost of a slight reduction in remanence [2]. However, the toll on remanence with the addition of Pr-Cu alloys is comparatively less prominent as the Dy-Cu system, since the antiferromagnetic coupling of substituting the Dy atoms in the hard-ferromagnetic phases causes a more noticeable drop in the net magnetization [46], which is apparent from Figure 1. Comparatively the *H_Ci_* improved from 1160 kA/m (RMs) to 1260 kA/m (900 °C GBDP) and 1308 kA/m (after secondary annealing) for 10 wt. % Pr-Cu diffusion processing (3.87% mass increase). Likewise, utilizing 20 wt. % (4.96% actual mass gain) Pr-Cu alloy ribbons as the direct diffusion source resulted in *H_Ci_* enhancement to 1290 kA/m (just after primary diffusion treatment—900 °C) and subsequently to 1322 kA/m with additional annealing (at 500 °C), as disclosed in Figure 2a. This approx. 5% mass gain with Pr-Cu diffusion treatment suggests 15% higher coercivity over the starting RM and 59.3% over the original recycled HDDR Nd-Fe-B powder (RP). These attained values are slightly better than the recently reported *H_Ci_* improvement by Pr-Cu diffusion processing of the hot deformed magnets to 1232 kA/m from the fresh HDDR Nd-Fe-B powders, with starting *H_Ci_* = 1065 kA/m [43].

The microstructural analysis was performed on 2 wt. % Pr-Cu diffusion processed RM after secondary annealing, using the backscattered electron imaging as shown in Figure 3. For the sake of comparison, the recycled HDDR powder (RP) and the reprocessed magnet (RM) prior to the GBDP with Pr-Cu alloys are also shown in Figure 3A,B respectively. Fractography of the HDDR powder particles will most inevitably reveal microstructure similar to Figure 3A, i.e. 3D network of Nd_2_Fe_14_B grains with preferred orientation along the easy-axis (c-axis) in each particle. The disproportionation of microcrystalline Nd_2_Fe_14_B matrix in the EOL magnets spreads uniformly at elevated temperatures to a reaction mixture of NdH_2_, α-Fe and Fe_2_B throughout the whole particle, such that the vacuum desorption of H_2_ and recombination reaction creates a transcending continuous 3D network of submicron sized Nd_2_Fe_14_B grains within each HDDR particle [2]. As reported previously, the recycled HDDR Nd-Fe-B powder contains a high oxygen content (~5000 ppm) and therefore formation of additional oxide phases (fcc-NdO_X_ and cubic/hcp Nd_2_O_3_—depending on oxygen concentration/up-take) reduces the overall amount of Nd-rich phase present within the system below 13.4 at. % [21,23]. The SPS reprocessed magnet (Figure 3B), has been developed with microstructure optimally retained conceivably as close as possible to the starting recycled powder and enhanced magnetic properties followed by annealing. The anticipated changes with GBD treatment are illustrated in Figure 3C–F for 2 wt. % Pr-Cu surface diffusion after the primary and secondary annealing. This specimen was also selected for the reason that nearly all the Pr-Cu alloy apparently diffused inside the magnet indicated by 2% weight gain. The BSE images indicates different zones of diffusion (bright phase indicates rare-earth rich phases) detailed in Figure 3C. The diffusion depth illustrated in this image is clearly limited as the Pr-Cu-rich species were located in near surface regions, ranging up to approximately 250 µm from the surface. The possible origin of extended diffusion zone must be associated with the primary GBD treatment at 900 °C, such that it is anticipated that time and concentration dependent diffusion happened with the liquification of aggregate RE-phase above the ternary transition temperature (665 °C), which enhanced the diffusion depth to ~600 µm from the surface. This lack of diffusion depth beyond ~600 µm can be assumed because of high surface tension caused by the eutectic melt-pool, which requires multiple micron sized channels to diffuse through the surface regions; besides time and concentration gradient limitations at the surface. As previously demonstrated, the HDDR Nd-Fe-B system is very much unlike the melt-spun ribbon flakes and the sintered magnets with several micron sized grains having thicker grain boundary channels, such that the RE-rich melt is transported thermodynamically by grain boundary channels. In this HDDR Nd-Fe-B system, the diffusive transport happens via inwards capillary suction along very thin grain boundary channels of ~3 nm thickness [44]. This complicates the diffusivity and the prodigious surface tension caused by a high mass fraction of eutectic species in nearly dense magnets require a greater number of surface perforations to penetrate inwards. Thinner channels in the HDDR Nd-Fe-B surges the required capillary forces for diffusivity towards the grain boundaries connecting the nanocrystalline grains. The perforations in the conventional sintered magnets from microcrystalline precursors are considerably thicker to promote grain boundary diffusion. The HDDR particle size is ~220 µm, whereas in the sintered magnets the particle size is around 5–10 µm, which indicates numerous grain boundary channels in the sintered magnets as compared to the particle boundaries accessible at the surface of HDDR Nd-Fe-B based magnets. The nanoscopic grain boundary channels connected to the particle boundaries in the HDDR Nd-Fe-B system are much thinner than the sintered magnets, in which diffusion is controlled by concentration gradient [46,55]. The first hindrance for the RE-rich liquid phase are the surface channels to diffuse the liquid phase along the particle boundaries, which in turn distribute this liquid phase along the grain boundaries under capillary diffusion [44]. Hence the effective particle boundary surface area in the HDDR Nd-Fe-B system is considerably lower than the sintered magnets, which is required to accelerate the diffusion from the surface region under capillary forces and so the diffusion depth is lower than 1 mm. Another factor to account for the partial diffusivity is related to preferential flow kinetics with the magnetization easy axis (c-axis) in the HDDR Nd-Fe-B system [23]. In case of GBD processing of bulk magnets, the powder particles are joined up with interparticle boundaries, such that for these regions oriented with (parallel to) the easy axis (c-axis) will tend to preferentially allow capillary diffusion in to the grain boundary channels adjoining the ultrafine Nd_2_Fe_14_B grains within the diffusion (Figure 3D) and extended zones (Figure 3E), in the vicinity of Pr-Cu rich pools at the intergranular junctions. The presence of darker features 100–200 µm in size from Figure 3C are merely powder particles misoriented during the SPS reprocessing under applied pressures [22].

The Pr-Cu diffusion treatment has already been reported that the Pr-rich liquid/intergranular phase induces strains in the Nd_2_Fe_14_B matrix phase which may also lead to strain-induced transformation of Nd_2_O_3_ above ternary point facilitated by Cu segregation [58]. Therefore, the secondary annealing at 500 °C helps in relaxing the microstructure with reduced defect density by smoothening the grain boundaries and reprecipitated (Pr,Nd)_2_Fe_14_B surfaces from the RE-enriched melt, which is beneficial for the improvement in the magnetic properties. Close-up image of diffusion zone is shown in Figure 3D, which indicates the various phases randomly distributed within the microstructure and quantified in Table 2. This region is concentrated with Pr-rich phases, with a high-volume fraction of bright regions corresponding to (Pr,Nd)O_X_ type species intercalated at the HDDR particle boundaries and the Nd_2_Fe_14_B intergranular regions. In the near surface regions up to extended diffusion zone and close to the Pr-rich phase, the stoichiometric composition of the matrix phase resembled (Pr,Nd)_2_Fe_14_B [28]. The surface of 2:14:1 grains partially melt during GBDP above ternary transition temperature in equilibrium with RE-rich phase, such that enrichment of facet region causes reprecipitation and asymmetrical solidification of shell structure at the interface (in case of HREEs) [55]. Previously it has been suggested that the grains facets in contact with (Nd,Pr)-rich liquid phase transformed to (Pr,Nd)_2_Fe_14_B due to Nd depletion but did not completely form the Pr-type shell structure over the Nd_2_Fe_14_B grains [58]. The EDXS results provide evidence for this transformation of nanocrystalline matrix grains during GBDP, such that the surfaces connected to the Pr-enriched liquid phase partially substitute Nd with Pr to form (Pr,Nd)_2_Fe_14_B facets and depleted Nd becomes part of liquid phase. In the diffusion region, the Pr-rich phase was analyzed with a nominal composition of (Pr,Nd)O_2_, besides the presence of slightly greyish Pr_2_O_3_/Nd_2_O_3_ oxide phase which is formed due to excessive availability of oxygen within the microstructure. Overall, the distribution of Pr, Nd, Cu and O at the intergranular channels with matrix phase and the particle boundaries is non-uniform and varies inconsistently with the diffusion depth. Furthermore it can also be hypothesized that the secondary annealing may not have contributed significantly in increasing the diffusion depth of RE-rich species, since the Pr-Cu alloy should have transformed to different phases as indicated in Figure 3D above the ternary transition temperature during the primary GBDP.

The miscibility of eutectic species in the Nd-rich intergranular phase is significantly higher and therefore, for the surface species to be propelled inside, rate of mass transport (diffusivity) is limited by concentration gradient which is also limited by time. Hence during the primary GBD treatment, it is anticipated that the diffusivity limits the penetration of melt to near surface regions only (up to 500–600 µm). Since the primary GBDP occurs above the ternary eutectic point (665 °C), so the Nd-rich phase is in liquid state with nanocrystalline Nd_2_Fe_14_B and Nd_1+ε_Fe_4_B_4_ phases [22]. Now the influx and dissolution of Pr-Cu will be limited by the concentration gradient of liquid phase along the HDDR particle boundaries before capillary infusion along the nanocrystalline grains. Initially Pr and Cu in the RE-rich melt become part of aggregate liquid phase at the particle boundaries making it enriched with the rare-earth species. As the capillary diffusion begins, Cu remains segregated within the liquid phase while Pr substitutes partially on the facets in contact with RE-rich liquid phase [43]. This capillary diffusion from the surface to particle boundaries and finally to grain boundaries is both time and concentration dependent, although considerably slow as compared to the localized diffusion previously devised by DyF_3_ doping of the HDDR Nd-Fe-B system [44]. The diffusion zone, shown in Figure 3D also indicates the presence of secondary phases like Nd_2_O_3_ at the intergranular junctions restrict the capillary flow of liquid phase, either by limiting the liquidus phase to localized regions only and/or scavenging the rare earth elements to form more dhcp-RE_2_O_3_ type phases [22]. Since EDXS indicates a more common distribution of Nd_2_O_3_ phase as compared to Pr_2_O_3_, which implies that Pr preferably interacts more with the surface facets of adjacent Nd_2_Fe_14_B grains or remain associated within the grain boundary channels. The segregation of Cu in the intergranular region has been observed to rearrange the Nd atoms during annealing at 500 °C, resulting into transformation of hcp-Nd_2_O_3_ oxide to cubic-Nd_2_O_3_ phase [43,59]. This suggests that Cu segregation induced changes to the hcp-Nd_2_O_3_ phase at the intergranular region favors the transformation to cubic type crystal structure, which releases the strain introduced by hcp-Nd_2_O_3_ phase due to their significantly higher mismatch with the (Pr,Nd)_2_Fe_14_B matrix grains [21]. Furthermore, the transformation of cubic-Nd_2_O_3_ phase originates from the oxidation of metallic Nd and cubic-PrO_2_ type phase stabilized by Cu at the intergranular junctions [43]. Additionally, the RE-rich phase has been identified within the diffusion zone in RE-O_X_ form, suggesting the bright region is composed of (Pr,Nd)O_X_ phase. The more lamellar form factor indicates NdO_2_/NdO type phase in the vicinity to (Pr,Nd)_2_Fe_14_B matrix, while the continuous white region suggests combined NdO_2_ and (Pr,Nd)O_2_ phase. More elaborate analytical analysis with the transmission electron microscopy to understand the capillary diffusion limiting factors at the 2–3 nm scale of grain boundaries is recommended, since the FEG-SEM in terms of EDXS resolution can precisely classify only the intergranular junctions and particle boundaries of size ≥ 0.5 µm and distinguishing different crystal structures was not possible with the existing setup.

It is anticipated that the secondary annealing above the Pr-Cu binary eutectic point 472 °C only resulted in re-melting and redistribution of localized Pr-Cu phase from the particle and intergranular boundaries. It is speculated based on EDXS results that although Cu remains segregated within the liquid phase, Pr and Nd switch positions asymmetrically, such that per this matrix facets substitution implies liquid phase becoming richer with Nd instead of Pr. The melting point of eutectic Nd-Cu alloy is 520 °C [42], indicating limited capillary flow distance and redistribution of Nd-rich species. However, this temperature is effective in releasing the thermal processing strains within the magnets, which subsequently resulted into an improvement of the coercivity as well as the remanent magnetization [22].

The lamellar structure of intergranular phase is rich with Cu and rare-earth elements illustrated in Figure 3E, corresponds to RE_2_CuO_2_ [60] with brighter contrast, while a darker tone RE_X_CuO_4_ phase (cuprate with long-range magnetic order) was also identified within this extended region [61] as the RE-rich phase was diffused inwards. The bright lamellar (Nd,Pr)_2_CuO_2_ is widely distributed within the ~600 µm diffusion zone indicating conversion of the primary Nd-rich phase NdO_X_ to (Pr,Nd)O_2_ phase, which is a thermodynamic transformation precipitated by Cu at the intergranular junctions and the grain boundaries [43]. This region in the extended diffusion zone, shown in and Appendix A, illustrates the RE-rich region containing complex cuprates [the darker (Pr,Nd)CuO_4_ and grayish lamellar (Pr,Nd)_2_CuO_2_ phase] are of size 300–500 nm, co-existing with (Pr,Nd)O_X_ phase. The magnetic exchange effects originating from (Nd,Pr)CuO_4_ at the intergranular regions are out of scope for concurrent study. In the vicinity, Pr from the melt also partially substitutes Nd in the hard-ferromagnetic matrix by transforming to (Pr,Nd)_2_Fe_14_B phase [46,58]. Previous literature reported the segregation of Cu and transition metals (Fe, Co) at the intergranular phase along the interface with (Pr,Nd)_2_Fe_14_B nanocrystalline grains [58], which apparently is also true for the HDDR Nd-Fe-B system; however, this type of analysis is beyond the scope of utilized FEG-SEM and EDXS technique. The region beyond the diffusion zone shown in Figure 3F primarily consisted of microstructure similar to that of the bulk sintered magnet prior to GBDP, with exception of very limited Nd-rich phase redistribution in sparse regions only. This suggests that the overall diffusion depth of eutectic Pr-Cu alloys from the surface is limited to ~600 µm in the HDDR Nd-Fe-B based bulk magnets, prioritizing the requirement of localized grain boundary engineering instead of surface diffusion treatments.

### 4.3. Magfine MF-15P Grain Boundary Diffusion Treatment

The commercial grade MF-15P type HDDR Nd-Fe-B has been sparsely used in research but has been popular for the bonded magnet applications due to suitable magnetic properties at room temperature and up to 100 °C [24]. In order to compare the effectiveness of Pr-Cu diffusion treatment, the MFMs were produced with similar SPS and GBD processing conditions. The optimally SPS-ed and annealed MFMs have starting *H_Ci_* = 968 kA/m, *B_r_* = 1.07 T, *BH_max_* = 198 kJ/m^3^ and 99% relative density. In this case with MFMs, 10 and 20 wt. % of Pr-Cu alloy ribbons were added to the crucible to achieve optimal diffusion treatment results to compare with GBDP MRs, as shown in Figure 4. The GBDP with 10 wt. % Pr-Cu alloy ribbons improved the *H_Ci_* to 1206 kA/m (at primary diffusion treatment) and 1255 kA/m (secondary annealing). The *B_r_* reduced slightly to 0.96 T for 4.11% mass gain (0.113 g), as plotted in Figure 4b. Similarly, according to Figure 4b, the MFM sample diffusion treated with 20 wt. % Pr-Cu alloy gained mass by 0.171 g (6.37%), which resulted in *H_Ci_* boosted to 1269 kA/m (at 900 °C) and consequently, after 500 °C annealing to 1313 kA/m with *B_r_* = 1.04 T. This resistance to demagnetization i.e. coercivity is approx. 35.6% better than optimally SPS processed MFMs and a modest 11.3% better than the previously reported [24].

Figure 3C–F confirms gradient microstructure of the Pr-Cu GBDP HDD Nd-Fe-B magnet with surface regions experiencing magnetic hardening as compared to the inner segment. The magnetic properties of the bulk HDDR Nd-Fe-B magnets after GBDP with the eutectic Dy-Cu and Pr-Cu alloys are organized in Table 3. The results indicate that although the GBDP of the eutectic alloy ribbons is limited in mass gain and diffusion depth to near surface regions (~600 µm for Pr-Cu alloy) in the HDDR system, the Dy-Cu alloys are comparatively less effective in improving the coercivity as compared to Pr-Cu alloys, besides a more significant impact on the magnetization reduction (*B_r_* < 0.1 T) in case of former type alloys due to the HREEs and unreacted Dy at the boundary interfaces adds up to the reduction in *B_r_* as well as *H_Ci_*. Henceforth, the Dy-Cu alloys will always cause reduction in the *B_r_* for the diffusing Dy implying that the Dy-Cu alloy ribbons did not achieve adequate liquefaction as the Pr-Cu ribbons under the 900 °C GBDP conditions, owing to their high eutectic temperature which caused very limited *H_Ci_* improvement in RMs. The improvement achieved with eutectic Pr-Cu alloys is approx. 60% in *H_Ci_* as compared to Dy-Cu alloys at 55% in the recycled HDDR Nd-Fe-B system. This effect of Pr-Cu alloys excelling Dy-Cu has been previously identified in the commercial grade sintered magnets treated with same composition of eutectic alloys diffusion processed at 900 °C for 4 h, suggested the lower melting Pr-Cu alloy registered approx. 240 kA/m higher *H_Ci_* over Dy-Cu alloy, although the latter alloy developed core-shell structures [46]. The secondary annealing is responsible for the relaxation of thermal stresses and reformation of grain facets, such that the grain boundary distribution is more uniform among the grains after the secondary annealing. As a matter of fact, the diffusing Dy-rich species via surface GBDP were found less effective as compared to the HREEs original added to the composition because of unreacted Dy at the interface and inhomogeneous core-shell formation, which in turn increases the coercivity µ_O_H_C_ > 2 T [56]. Similarly, the Pr-Cu alloy ribbons enhanced the *H_Ci_* without negligible impact on the *B_r_* of the fresh MF-15P HDDR Nd-Fe-B system by 25% over the starting HDDR powder (MFP) and pre-GBDP bulk magnet (MFM) by 36% approximately. Evidently for the HDDR Nd-Fe-B system, the observed mass gain associated with Dy-Cu alloy is lower for all weight fractions as compared to Pr-Cu alloys confirming the thermally controlled concentration gradient effect here. This suggests that the eutectic Dy-Cu system may be more effective for the HDDR Nd-Fe-B powders instead of the bulk magnets subsequently due to rapid and uniform short-range diffusion over loose powder with a high surface area at these GBDP conditions. In case of the HDDR powders, there may not be a requirement to facilitate short-range diffusion from the particle interfaces, so GBDP HDDR powder with Dy-Cu alloys can later be compacted with SPS and annealed as a general suggestion for the future work. A pragmatic recommendation for future research would also be to mill-down the eutectic alloy ribbons to reduce the surface tension associated effects which limit the capillary diffusion and redistribution of RE-rich species under concentration gradients. Another alternative is to uniformly mix the finely milled alloy ribbons with the rare-earth-lean recycled HDDR Nd-Fe-B powder and subsequently sinter them with the SPS at even lower temperatures, such that overall processing steps are considerably reduced. Utilization of hyper-eutectic rare-earth rich alloys instead also has effectively served the purpose in the sintered and hot deformed magnets [45,46,47,56,58].

## 5. Conclusions

The HDDR Nd-Fe-B based bulk magnets were grain boundary diffusion (GBD) treated with the eutectic alloy ribbons of Pr-Cu and Dy-Cu in the range of 900 °C. The variation in magnetic properties was studied with respect to different weight fractions of Pr-Cu and Dy-Cu alloy ribbons placed on top and bottom of the bulk magnets for thermal processing. The GBDP resulted in improvement of the coercivity of the HDDR Nd-Fe-B systems, bulk magnets made from both fresh and recycled materials. The high temperature Dy-Cu eutectic was found less efficient as compared to low melting Pr-Cu binary eutectic during the scheme of primary processing at 900 °C and secondary annealing at 500 °C resulted in overall improvement *H_Ci_* by 60% at 1322 kA/m as compared to 830 kA/m for the recycled powder. Correspondingly, the fresh MF-15P HDDR Nd-Fe-B based bulk magnets gained up to 36% improvement in *H_Ci_* without a substantial decrease in *B_r_*, proving the suitability of GBDP with binary eutectic alloys. Therefore, the transport of liquid phase from the surface to the particle boundaries is postulated to occur under the effect of concentration gradient which is controlled by temperature and interparticle phase chemistry, whereas further from the particles to the grain boundaries is activated by capillary transport. On the contrary, the mass gain was lower than total diffusing species. It was further observed with SEM analysis that the diffusion treatment is limited in achieving thorough surface penetration, primarily because of reduced capillary forces from high weight fraction of molten species creating surface tension and lack of excessive particle boundaries in fully dense magnets. The matrix adjacent to the diffusion zone up to ~600 µm constitutes of (Pr,Nd)_2_Fe_14_B nanocrystalline grains with RE-rich (Pr,Nd)O_2_ and the secondary phases: lamellar Nd_2_CuO_2_ and (Nd,Pr)CuO_4_ at the intergranular regions. The region beyond the diffusion zone is similar to the bulk magnets prior to the GBD treatment. Nonetheless, the retention of magnetization (*B_r_*) after GBDP indicate limited surface diffusivity for the mass gain is still effective in revitalizing the coercivity and the bulk magnets have gradient microstructure with magnetic hardening of the surface regions. Convincingly, this report also suggests that eutectic Dy_70_Cu_30_ is not an optimal alloy for GBDP of the HDDR Nd-Fe-B bulk magnets because of the requirement for very high processing temperatures to activate the mass diffusion. The effectiveness of eutectic Dy-Cu alloy’s is substantially slower than Pr-Cu alloys based on mass and coercivity gain, thus limiting its GBDP applicability to HDDR Nd-Fe-B powders only, since processing above 900 °C may deteriorate the magnetic properties due to unnecessary abnormal grain coarsening.

## Figures and Tables

**Figure 1 materials-13-03528-f001:**
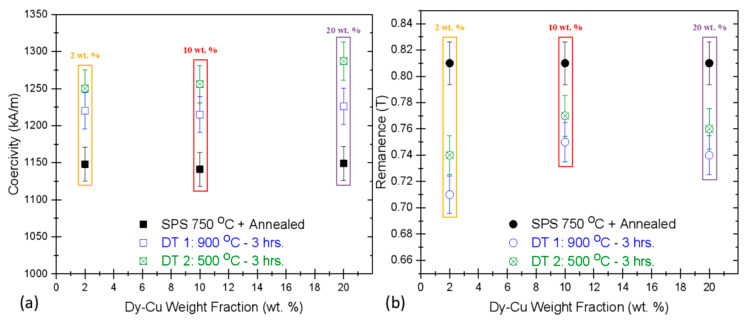
(**a**) Dependence of coercivity (*H_Ci_*) and (**b**) remanence (*B_r_*) with different weight fractions of eutectic Dy_70_Cu_30_ alloys during the two stages of GBD treatment. Plots legend: yellow 2 wt. %, red 10 wt. % and purple 20 wt. % Dy-Cu as the diffusion source on bulk SPS reprocessed magnets (RMs).

**Figure 2 materials-13-03528-f002:**
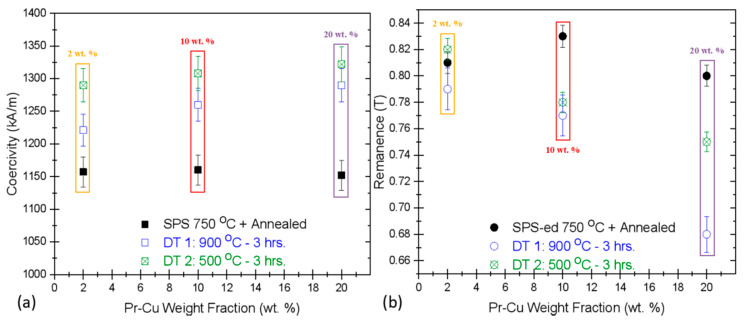
The variation of (**a**) coercivity (*H_Ci_*) and (**b**) remanence (*B_r_*) after GBD treatment with different wt. % Pr_68_Cu_32_ alloy ribbons. Plots legend: yellow 2 wt. %, red 10 wt. % and purple 20 wt. % Pr-Cu as the diffusion source on bulk HDDR Nd-Fe-B SPS-ed magnets (RM).

**Figure 3 materials-13-03528-f003:**
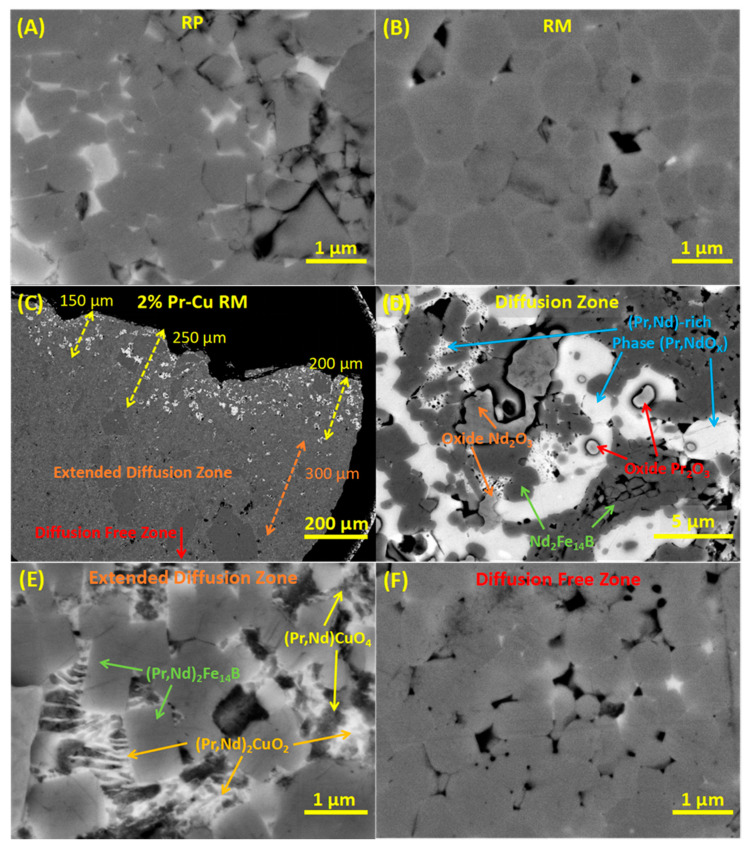
BSE-SEM analysis, high magnification image of: (**A**) bonded recycled HDDR Nd-Fe-B powder (RP); (**B**) optimally SPS-ed HDDR Nd-Fe-B bulk magnet (RM) prior to GBDP; (**C**) micrograph of 2 wt. % Pr-Cu GBDP specimen indicating diffusion zones after secondary annealing; (**D**) close-up image of diffusion zone with different phases quantified in Table 2; (**E**) the diffusion zone extension due to secondary annealing showing the distribution of Pr-Cu-rich phase at the intergranular regions; and (**F**) diffusion free zone which mimics the microstructure of original RM due to limited diffusion depth of Pr-Cu alloys in HDDR Nd-Fe-B system.

**Figure 4 materials-13-03528-f004:**
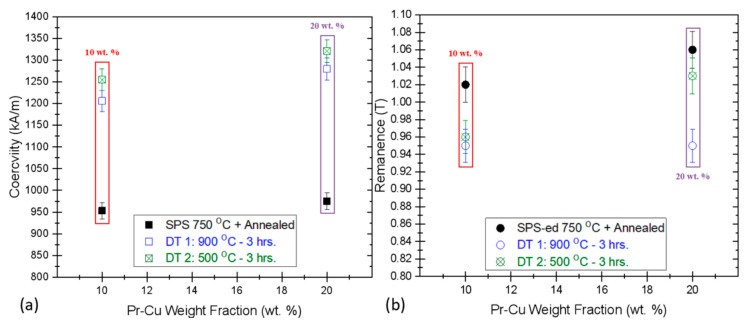
Illustrates the deviations in (**a**) coercivity (*H_Ci_*) and (**b**) remanence (*B_r_*) with GBD treatment for different wt. % Pr_68_Cu_32_ alloy ribbons. Plots legend: red 10 wt. % and purple 20 wt. % Pr-Cu as the diffusion source on fresh HDDR Nd-Fe-B SPS-ed magnets (MFM).

**Table 1 materials-13-03528-t001:** Magnetic properties of HDDR Nd-Fe-B magnets prior to Pr-Cu and Dy-Cu GBDP.

Material Class	Coercivity *H_Ci_* (kA/m)	Remanence *B_r_* (T)	*BH_max_* (kJ/m^3^)	*M_r_*/*M_S_* Ratio
End-of-Life (EOL) Scrap Magnet	1170	1.19	250	0.74
Recycled HDDR Nd-Fe-B Powder (RP)	830	0.9	124	0.56
Optimally SPS-ed and Annealed Recycled Magnets (RMs)	1150–1170	0.79–0.83	112–120	0.52
Fresh Magfine MF-15P HDDR Nd-Fe-B Powder (MFP)	1020–1130	1.27–1.32	270–310	>0.81
Optimally SPS-ed and Annealed Fresh MF-15P Magnets (MFMs), see Section 4.3	960–970	1.06–1.07	196–200	0.67

**Table 2 materials-13-03528-t002:** FEG-SEM EDXS quantification of different phases in the 2 wt. % Pr-Cu GBD processed recycled HDDR Nd-Fe-B bulk magnets.

Phases	Nd (at. %)	Pr (at. %)	Dy (at. %)	Fe (at. %)	O (at. %)	Al (at. %)	Cu (at. %)
Nd_2_Fe_14_B	13.1	2.2	–	80.7	–	1.1	2.9
(Pr,Nd)_2_Fe_14_B	4.2	9.3	–	85.6	–	0.9	–
(Pr,Nd)-rich Phase (Pr,Nd)O_x_/(Pr,Nd)O_2_	7.5	19.5	–	1.5	71.5	–	–
Pr_2_O_3_ /Nd_2_O_3_	10.4	15.4	–	10.5	63.7	–	–
(Nd,Pr)_2_CuO_2_	20.1	13.4	2.1	14.8	30.9	–	18.7
(Pr,Nd)_X_CuO_4_	9.1	13.0	–	3.0	51.4	–	23.5
RE_2_CuO_2_	23.7	16.5	3.3	10.7	32.6	–	13.2

* Oxford Instruments INCA 350 EDXS 20 mm^2^ detector Point ID analysis system at 20 kV and XPP matrix corrections with quantitative error for light elements in standard deviation (S.D) ~0.085 and for combined elements S.D ~0.055. Quant and Profile optimization applied for normalized Point ID analysis with >10 k cps during all individual quantitative measurements. All individual measurements except the last three were repeated twice and average at. % values are reported in this study.

**Table 3 materials-13-03528-t003:** Magnetic properties of HDDR Nd-Fe-B bulk magnets after Pr-Cu and Dy-Cu GBDP.

	Diffusion Source Alloy Weight Fraction	Coercivity H_Ci_ (kA/m)	Remanence B_r_ (T)	BH_max_ (kJ/m^3^)	Actual Mass Gain (%)
Dy_70_Cu_30_ + Recycled Magnet (MR)	0 wt. %	1148	0.81	115	–
	2 wt. %	1250	0.74	90	1.94
	10 wt. %	1256	0.78	105	2.57
	20 wt. %	1287	0.77	104	3.25
Pr_68_Cu_32_ + Recycled Magnet (MR)	0 wt. %	1160	0.83	120	–
	2 wt. %	1283	0.82	116	2.00
	10 wt. %	1308	0.78	107	3.87
	20 wt. %	1322	0.75	100	4.96
Pr_68_Cu_32_ + Fresh Magnet (MFM)	0 wt. %	969	1.07	198	–
	10 wt. %	1255	0.96	127	4.11
	20 wt. %	1313	1.04	190	6.37

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
