# Peer review of "Limitations in the Grain Boundary Processing of the Recycled HDDR Nd-Fe-B System"

_materials, 2020, doi:10.3390/ma13163528_

Round 1
Reviewer 1 Report
The rudimentary aim of this scientific briefing is to highlight the potential challenges in grain boundary diffusion processing (GBDP) of the recycled HDDR Nd-Fe-B, but the potential challenges mentioned in the introduction are not clearly indicated.
Regarding the result, the data in this paper is not sufficient. For example, the SEM diagram of the Dy-Cu GBDP samples of recycled HDDR Nd-Fe-B and GBDP samples of MF-15P HDDR Nd-Fe-B are missing. In addition, the magnetic performance data of the MF-15P HDDR Nd-Fe-B with 2% Pr-Cu are still missing.
About the discussion, the existing experimental results have not been fully discussed. For example, there is no relevant data to explain the reason why Dy-Cu GBDP samples are found less efficient as compared to Pr-Cu, and why the coercivity of Fresh Magnet samples are lower than Recycled Magnet samples.
Finally, there are some format errors. The serial numbers of the subheadings on lines 123, 169 and 347 are wrong, and there is a question about the parallel relationship of these subtitles. And there is a problem with the description of nonferromagnetic grain boundaries in lines 46 and 47. Besides, the number of lines 83 and later is not marked in terms of alloy composition is the atomic number ratio rather than the mass ratio.
Author Response
Thank you for the review. Please find enclosed our response to the queries posted by the reviewer in the file attached.
Kind Regards

Reviewer 2 Report
The submitted manuscript entitled ‘Limitations in the grain boundary processing of the 2 recycled HDDR Nd-Fe-B system”. The structure of the scientific report is good and well-understood. The aim is clarified. The introduction summarizes in order to fill this literature gap.
Discussion:
“The concurrent research work addresses the diffusion mechanism in the recycled HDDR Nd-Fe-B system, which has never been reported before. The details are presented in context of mass gain with diffusion processing parameters and the diffusion depth limitations in the system due to limited capillary channeling and presence of intergranular oxides with the aid of SEM and EDXS analysis.
This report also indicates further action plans for the future on GBDP of the HDDR Nd-Fe-B system to tackle the limitations in diffusion depth and augment the coercivity beyond the current state of the art values.”
Bibliographic references are appropriate. The author mentioned 66 literatures according to the reference list. These references are from the last 10 years, from prestigious journals.
The manuscript is interesting, during its review only a technicalities arose.
-error limits are missing from the table 2.
The reviewer suggests to accept after minor revision form in the paper for publication.
Author Response

(The authors gave the same response as above.)

Reviewer 3 Report
The current article seems to be interesting and may attract wide readers. However, need some clarifications.
- On page 3 line 118, the authors have written ‘… annealed bulk magnets measured before the GBD treatment.’ Is this measurement did at room temperature?
- On page 4 line 134, authors have mentioned ‘… for 2 wt. % sample approx. 0.019 g (1.94 %), 10 wt. % sample approx. 0.068 g (2.57 %) and 20 wt. % sample approx. 0.0911 g (3.25 %)’, what are the values in the bracket, 1.94 %, 2.57%, 3.25% ?
- Authors have mentioned on page 4 that, secondary heating helped slightly recover the Remanence, Is this mean if you do a third time heating you can still recover Remanence?
- If you analyze Figure 1a carefully, a major change in coercivity is observed only for 20 wt% sample after secondary heating, what is so special about this?
- If you see Figure 2b, there is no specific trend or pattern for the change in Br. For 10 wt%, the Br for SPS increased to 0.83T whereas for 20 wt% it reduced to 0.80 T, What could be the reason for this fluctuation in Br?
- On page 6 the authors have mentioned that their values are slightly better than the recently reported Hci values. What is the reason for this improvement?
- By looking at the Coercivity and Remanence graph, is it possible to say that the variation in the values is due to GBD?
- Need some clarification concerning the microstructure provided in Figure 3, authors have mentioned that these are the cross-sectional images. Are Figures 3A and B also Cross-sectional images?
In Figure 3C authors identified diffusion zone, extended zone, and diffusion free zone, in such a low magnification how can we see whether there is diffusion? The white particles or agglomerates in Figure 3C looks like particles deposited during cutting the substrate for a cross-section image. It looks like while cutting the substrate for cross-section image a lot of cracks formed and some particles might have gone into that crack also. Pd, Nd rich region shown in Figure 3D is very porous, what is the reason for this porosity.
Concerning the scale bar given in Figure 3D, it is a little unrealistic to see the grain boundary diffusion.
In Figure 3E, more than diffusion it looks like some defects.
Is it possible to calculate quantitatively the amount of Material diffuse through grain boundary?
The microstructures showed in the manuscript is for 2 wt%, is the qualitative GBD varies for other wt%? Is this variation correlated with magnetic properties?
Need magnified images to see the grain boundaries more clearly.
The manuscript needs typo and grammatical correction.
Author Response

(The authors gave the same response as above.)

Round 2
Reviewer 1 Report
Significant improvement has been made in the revised manuscript. It is suggested to be published in the MDPI Materials.
Author Response
We would like to thank the reviewer for critical suggestions in terms of scientific clarity and soundness of the message to the readers. These helpful comments have to be credited for the improvement in the quality and legibility of the manuscript.
Lastly, we completely reassure that the manuscript has been revised for spelling and grammatical mistakes with multiple authors proofreading the text before resubmission to the editor. We hope no lousy mistakes will be further identifiable during the review.
We look forward to a promising discussion or any further comments from the reviewer.
Warmest Regards
Reviewer 3 Report
Appreciate authors for taking time and answering most of my questions. I am convinced and happy with the replies made by authors. But, I am not still convinced by the BSE-SEM images.
Authors claim these are the cross-sectional images, but the images look like a planar top view for me. I would imagine/expect layers of different zones in a cross-sectional image, which is missing in all the cross-sectional images in Figure 3. May be it is just the matter of rotation of images. It will be good, if the authors can modify the images in such a way that, they looks like cross sectional images.
Author Response
Please find attached our detailed answer to the reviewer's queries.
Thank you and kind regards
